# Rad1 and Rad10 Tied to Photolyase Regulators Protect Insecticidal Fungal Cells from Solar UV Damage by Photoreactivation

**DOI:** 10.3390/jof8111124

**Published:** 2022-10-25

**Authors:** Lei Yu, Si-Yuan Xu, Xin-Cheng Luo, Sheng-Hua Ying, Ming-Guang Feng

**Affiliations:** Institute of Microbiology, Collegeof Life Sciences, Zhejiang University, Hangzhou 310058, China

**Keywords:** filamentous fungi, anti-UV proteins, photolyase regulator, DNA photorepair, photoreactivation, nucleotide excision repair

## Abstract

*Beauveria bassiana* serves as a main source of global fungal insecticides, which are based on the active ingredient of formulated conidia vulnerable to solar ultraviolet (UV) irradiation and restrained for all-weather application in green agriculture. The anti-UV proteins Rad1 and Rad10 are required for the nucleotide excision repair (NER) of UV-injured DNA in model yeast, but their anti-UV roles remain rarely exploredin filamentous fungi. Here, Rad1 and Rad10 orthologues that accumulated more in the nuclei than the cytoplasm of *B. bassiana* proved capable of reactivating UVB-impaired or UVB-inactivated conidia efficiently by 5h light exposure but incapable of doing so by 24 h dark incubation (NER) if the accumulated UVB irradiation was lethal. Each orthologue was found interacting with the other and two white collar proteins (WC1 and WC2), which proved to be regulators of two photolyases (Phr1 and Phr2) and individually more efficient in the photorepair of UVB-induced DNA lesions than either photolyase alone. The fungal photoreactivation activity was more or far more compromised when the protein–protein interactions were abolished in the absence of Rad1 or Rad10 than when either Phr1 or Phr2 lost function. The detected protein–protein interactions suggest direct links of either Rad1 or Rad10 to two photolyase regulators. In *B. bassiana*, therefore, Rad1 and Rad10 tied to the photolyase regulators have high activities in the photoprotection of formulated conidia from solar UV damage but insufficient NER activities in the field, where night (dark) time is too short, and no other roles in the fungal lifecycle in vitro and in vivo.

## 1. Introduction

Strong UVB (280–320 nm) and UVA (320–400 nm) irradiations in summer sunlight [1] are ubiquitous outdoor stresses that restrain all-weather applications of fungal insecticides and acaricides to protect green agriculture from arthropod pest damage [2,3,4,5]. As common active ingredients of such pesticides, fungal conidia are more vulnerable to the damage of UVB than of UVA [6,7]. The damage arises from an impairment of macromolecules in UVB-absorbed cells and an exposure of fungal cells to the oxidative stress of reactive oxygen species generated by UVA irradiation [8,9]. Thus, understanding the roles and mechanisms behind fungal anti-UV is critical for the development of an optimal application strategy enabling the protection of formulated conidia from solar UV damage and the enhancement of fungal pest control efficacy [10].

As a vital macromolecule, eukaryotic DNA is readily damaged by UV irradiation, which induces the formation of two distinctive cytotoxic photoproducts known as cyclobutane pyrimidine dimer (CPD) and (6-4)-pyrimidine-pyrimidone (6-4PP). The UV-induced CPD and 6-4PP DNA lesions harmful to cells [11,12,13] form through covalent linkages between adjacent bases of the DNA duplex [14], and their repair relies on two distinctive mechanisms [15]. Shorter UV-induced DNA lesions can be rapidly repaired by exposure to longer UV or visible light. The rapid repair process known as photorepair relies on light energy for the direct transfer of electrons to CPD or 6-4PP lesions to break down the covalent linkages under the actions of photolyases [16,17,18]. Filamentous fungi possess only one or two photolyases, i.e., Phr1 specific to CPD and/or Phr2 specific to 6-4PP [19]. In filamentous fungi, DNA lesions are photorepaired by Phr1 and/or Phr2 rather than one or two DASH-type cryptochromes (Cry-DASHs) classified in the same cryptochrome-photolyase family [20,21,22,23], which comprises one to four members [19]. In some mucoromycetous fungi, Cry-DASH (CryA),the unique member of that family, has been reported to repair CPD lesions in single- and double-stranded DNAs in vitro and enable photoreactivation in vivo [24,25], a viability recovery process of UV-impaired cells under light. Indeed, mucoromycetous CryA is identical to Phr1 in domain architecture, and its photorepair activity seems to be associated with a nuclear localization signal (NLS) motif that is shared by Phr1 or Phr2 but not by Cry-DASHs in non-mucoromycetous fungi [19], although its nuclear localization was not disclosed in previous studies. These studies indicate that filamentous fungal DNA photorepair is dependent on nuclear photolyases but independent of cytosolic Cry-DASHs.

Nucleotide excision repair (NER), the other DNA repair mechanism studied intensively in *Saccharomyces cerevisiae* [15,26], proceeds slowly in darkness but has been rarely explored in filamentous fungi. Unlike filamentous fungal photorepair depending on one or two photolyases, the budding yeast’s NER relies on a large array of enzymes and proteins involved in proteasome activity and poly-ubiquitination in the dark [27,28,29,30]. Anti-UV genes involved in the yeast cells’ surviving UV radiation (RAD genes) were identified in early studies [31,32]. Indeed, a large family of RAD proteins and associated partners have been classified as belonging to the NER pathway, and multiple RAD-RAD and RAD-containing complexes are evidently required for the NER processes, including the recognition, opening, incision, and/or repair of DNA lesions [27]. Of those, a complex formed by Rad1 and Rad10 interacting with each other has been identified as an endonuclease enabling the recognition of the junction between single- and double-stranded DNAs and specifically the removal of unpaired 3′ tails by nicking within the duplex DNA from the junction [33,34,35]. This endonuclease is essential for the dual incision of UV-induced DNA lesions [36] and the microhomology-mediated end-joining and single-strand annealing [37,38], and it can remove 3′ dirty ends and nonhomologous 3′ tails generated during base excision repair and homologous recombination in *S**. cerevisiae*, respectively [39,40,41,42]. However, the anti-UV roles of Rad1 and Rad10 have been rarely studied in filamentous fungi. As a unique example, the Rad1 orthologue identified in *Neurospora crassa* was shown to be essential for UV resistance and considered to have a UV-injured DNA excision activity [43].

Notably, UVC (<280 nm), a UV component removed by atmospheric ozone from sunlight [1], was often used in the previous studies on filamentous fungal photorepair or the budding yeast’s NER. Applied fungal pesticides are usually exposed to solar UVB as a main source of DNA lesions to be repaired by photolyases [44] but unlikely to be exposed to the most harmful UVC that cannot reach the Earth’s surface. Interestingly, proteins homologous to most of those anti-UV proteins identified in the budding yeast exist in *Beauveria bassiana*, which serves as a main source of wide-spectrum fungal pesticides [5]. In *B. bassiana*, a protein orthologous to the yeast NER-required Rad23 [29,45,46,47] has been shown to photoreactivate UVB-inactivated conidia by its interaction with Phr2 but to not reactivate those conidia via a 24 h dark incubation [48]. In *Metarhizium*
*robertsii*, the white collar proteins WC1 and WC2 deficient of any DNA_photolyase domain required for photorepair activity have proved able to photorepair 6-4PP and CPD DNA lesions, respectively, for the photoreactivation of UVB-inactivated conidia [49].This is because, as proven in the previous study, either WC1 or WC2 interacts with two photolyases and regulates the expressions of most RAD genes homologous to those involved in the yeast NER. Filamentous fungal WC1 and WC2 usually interact with each other to form a white collar complex (WCC), which acts as a core regulator of light-responsive genes involved in transcriptional regulation, chromatin rem+odeling, and posttranslational modifications [18] and is also required for the precise timekeeping of the circadian clock in *N**. crassa* [50,51] and *M. robertsii* [52]. A circadian day of summer features longer daytime than nighttime, which is too short for sufficient NER in the field. Recent findings have led to a hypothesis that photorepair may serve as a main mechanism of filamentous fungal resistance to solar UV and depend on the WCC-cored pathway consisting of not only photolyases but multiple RAD proteins [19]. This study seeks to test the hypothesis by the characterization of orthologous Rad1 and Rad10 in *B. bassiana*. Our results confirm the interactions of either Rad1 or Rad10 with both WC1 and WC2, which are proven to act as photolyase regulators, and their high activities in photoreactivation but an insufficient role of each in NER.

## 2. Materials and Methods

### 2.1. Microbial Strains and Culture Conditions

The wild-type strain *B. bassiana* ARSEF 2860 (WT hereafter) was incubated on Sabouraud dextrose agar (4% glucose, 1% peptone, and 1.5% agar) plus 1% yeast extract (SDAY) at the optimal regime of 25 °C and a 12:12 (L:D) photoperiod and used as a recipient of targeted gene manipulation. The minimal medium Czapek–Dox agar (CDA: 3% sucrose, 0.3% NaNO_3_, 0.1% K_2_HPO_4_, 0.05% KCl, 0.05% MgSO_4_, and 0.001% FeSO_4_ plus 1.5% agar) was used to assay fungal responses to different types of chemical stressors at 25 °C. *Escherichia coli* DH5α and TOP10 incubated in Luria–Bertani medium at 37 °C were used for vector propagation. *Agrobacterium tumefaciens* AGL1 incubated at 28 °C was used for mediation of fungal transformation. The *S. cerevisiae* Y1HGold, Y187, and Y2HGold were used for yeast one-hybrid (Y1H) and two-hybrid (Y2H) assays.

### 2.2. Recognition and Bioinformatic Analysis of Rad1 and Rad10 Orthologues

The *S. cerevisiae* Rad1 (NP_015303) and Rad10 (NP_013614) sequences were used as queries to identify orthologues in the NCBI databases of selected fungi by BPLASTp analysis (http://blast.ncbi.nlm.nih.gov/blast.cgi, accessed on 3 October 2022). The queries and the resultant orthologues were subjected to sequence alignment analysis and clustered with a maximum likelihood method in MEGA7 (http://www.megasoftware.net/, accessed on 3 October 2022). Conserved domains and NLS motif predictedfrom either orthologue at http://smart.embl-heidelberg.de/ (accessed on 3 October 2022) and http://nls-mapper.iab.keio.ac.jp/ (accessed on 3 October 2022) were compared between *B. bassiana* and *S. cerevisiae*.

### 2.3. Subcellular Localization of Rad1 and Rad10

The green fluorescence protein (GFP)-tagged Rad1 and Rad10 fusion proteins were expressed in WT using the backbone vector pAN52-gfp-bar driven by P*tef1*, an endogenous promoter [44]. The open reading frame (ORF) of *rad1* or *rad10* was cloned from the WT cDNA and inserted into the 5′-terminus of *gfp* (U55763) in the linearized vector using a one-step cloning kit (Vazyme, Nanjin, China). The resultant vector was integrated into WT by means of *Agrobacterium*-mediated transformation and screened by the *bar* resistance to phosphinothricin (200 μg/mL). A desirable transformant (showing strong green fluorescence) from each transformation was incubated for conidiation on SDAY. Its conidia were suspended in SDBY (SDAY free of agar) for a 3-day shaking incubation at 25 °C and L:D 12:12. Hyphal samples from the cultures were stained with 4.16 mM DAPI (4′,6′-diamidine-2′-phenylindole dihydrochloride; Sigma-Aldrich, Shanghai, China) and visualized with laser scanning confocal microscopy (LSCM) at the excitation/emission wavelengths of 358/460 and 488/507 nm to determine subcellular localization of Rad1-GFP and Rad10-GFP. Green fluorescence intensity was assessed from a fixed circular area moving in the cytoplasm and nucleus of each of 15 hyphal cells using ImageJ software (https://imagej.nih.gov/ij/, accessed on 3 October 2022) to compute the nuclear versus cytoplasmic green fluorescence intensity (N/C-GFI) ratio as the nuclear accumulation level of Rad1-GFP or Rad10-GFP.

The vector pAN52-mCherry-sur [53] was modified with P*tef1* for co-localization of Rad1 and Rad10. The *rad10* ORF was ligated to the 5′-terminus of *mCherry* (KC294599) in the modified vector, followed by transformation into the strain expressing Rad1-GFP. Transgenic colonies were screened by *sur* resistance to chlorimuron ethyl (10 μg/mL). A colony exhibiting strong red fluorescence was chosen for co-localization of Rad1-GFP and Rad10-mCherry by LSCM at the excitation/emission wavelengths of 488/507 and 561/610 nm, respectively.

### 2.4. Yeast Hybrid Assays

Y2H assays were performed to detect protein–protein interactions among Rad1, Rad10, Phr1, Phr2, WC1, andWC2, as described previously [49]. Briefly, the ORFs of *rad1* (BBA_07749), *rad10* (BBA_03417), *wc1* (BBA_10271), *wc2* (BBA_01403), *phr1* (BBA_01664), and *phr2* (BBA_01034) were amplified from the WT cDNA with paired primers (Appendix A) and inserted into the prey vector pGADT7 (AD) or the bait vector pGBKT7 (BD). The constructs verified by sequencing were transformed into *S. cerevisiae* Y187 and Y2HGold, respectively, for 24 h pairwise yeast mating at 30 °C on YPD (1% yeast extract, 2% peptone, 2% glucose plus 0.04% adenine hemisulfate salt). The generated diploids were screened in parallel with positive (AD-LargeT-BD-P53) and negative controls (AD-BD and AD-*x*-BD or AD-BD-*x* (*x*: each target protein) on a synthetically defined medium (SDM) deficient of two (SDM/-Leu/-Trp/X-α- Gal/AbA, double dropout) and four nutrients (SDM/-Leu/-Trp/-Ade-His/X-α-Gal/AbA, quadruple dropout), respectively. Yeast colonies initiated with 500 to 50,000 cells were incubated at 30 °C for 3 days.

The activity of WC1 or WC2 binding to the promoter region of *phr1* or *phr2* was detected in the Y1H assays based on Matchmaker^®^ Gold Yeast One-Hybrid System (TaKaRa, CA, USA). Briefly, DNA fragments of P*phr1* (1829 bp) and P*phr2* (1904 bp) were amplified from the WT DNA and ligated to SmaI site in pAbAi using ClonExpress II One Step Cloning Kit (Vazyme, Nanjing, China). The constructs pAbAi-P*phr1* and pAbAi-P*phr2* were integrated into *S. cerevisiae* Y1HGold by homologous recombination, yielding bait- specific reporter strains that were screened on a selective uridine-dropout (SD/-Ura) medium. Further, the amplified *wc1* or *wc2* ORF was ligated to pGADT7 at EcoR1/BamH1 sites using the same kit. The resultant pGADT7-*wc1* and pGADT7-*wc2* were transformed into the bait-specific reporter strains, respectively. The protein–DNA interaction was determined by the yeast constructs grown for 3 days at 28 °C on a synthetic dextrose medium deficient of uridine and leucine but supplemented with aureobasidin A of 500 ng/mL (SD/-Leu+AbA). The strains transformed with pGADT7-*p53* and pAbAi-P*p53* were used as positive controls while the strains transformed with one or two empty vectors were used as negative controls. Yeast colonies initiated with 30 to 3000 cells were incubated at 30 °C for 3 days.

### 2.5. Construction and Identification of Targeted Gene Mutants

The deletion and complementation mutants of *rad1*, *rad10*, *wc1*, and *wc2* were generated following previous protocols [44,48,49]. A partial flanking/coding DNA fragment of each target gene was deleted by transforming the constructed vector p0380-5′*x*-bar-3′*x* (*x* = *rad1*, *rad10*, *wc1*, or *wc2*) into the WT strain, as mentioned, for homologous recombination of its 5′ and 3′ coding/flanking fragments (Appendix A). The p0380-*x*-sur vector (*x*, a full-coding sequence of each target gene with flank regions) constructed for targeted gene complementation was ectopically integrated into an identified Δ*x* mutant using the same transformation system. The *bar* or *sur* resistance, as mentioned, was exploited to screen putative Δ*x* or Δ*x::x* colonies. Expected recombination events were identified via PCR detection of DNA samples (Appendix A) and verified by real-time quantitative PCR (qPCR) analysis of cDNA samples (Appendix A). All paired primers used for amplification of DNA fragments and detection of each target gene are listed in Appendix A. The identified Δ*x* and Δ*x::x* mutants and the parental WT strain were used in the following experiments including three independent replicates each.

### 2.6. Assays for Growth Rates, Conidial Yields, Stress Tolerance, and Virulence

For each of the Δ*rad1* and Δ*rad10* mutants and their control (WT and complementation) strains, radial growth was initiated by spotting 1 μL aliquots of a 10^6^ conidia/mL suspension on SDAY, 1/4 SDAY (amended with 1/4 of each SDAY nutrient), CDA, and CDAs amended with different carbon or nitrogen sources. Incubated for 7 days at the optimal regime, each colony diameter was measured perpendicularly to each other across the center. The same method was used to initiate 7-day radial growth of each strain at 25 °C on CDA plates, which were supplemented with methyl methanesulfonate (325 μg/mL) or mitomycin C (33.4 μg/mL) for DNA-damaging stress, NaCl (0.7 M) or sorbitol (1 M) for osmotic stress, H_2_O_2_ (2 mM) or menadione (0.02 mM) for oxidative stress, and Congo red (3 μg/mL) or calcofluor white (10 μg/mL) for cell wall stress. In addition, spotted SDAY or CDA plates were incubated for 7 days after exposure to UVB irradiation at 0.1 J/cm^2^ (detailed in Section 2.7) and 2-day-old SDAY colonies initiated at 25 °C were exposed to a 42 °C heat shock for 3 to 9 h and then incubated at 25 °C for 5-day growth recovery, followed by estimation of colony diameters as mentioned. The measured diameters of stressed and unstressed (control) colonies were used to compute the relative growth inhibition percentage of each strain under each stress.

Conidial yield of each strain was assessed from each of three 7-day-old SDAY cultures initiated at the optimal regime by spreading 100 μL of a 10^7^ conidia/mLsuspension (the same below unless specified) per plate and standardized to the number of conidia per unit area (cm^2^) of plate culture. Conidial viability was assayed as median germination time (GT_50_) on agar plates at 25 °C.

Standardized bioassays on *Galleria mellonella* larvae (4th instar) were carried out by immersing three groups of 30–40 larvae per strain in 40 mL aliquots of conidial suspension for normal cuticle infection (NCI) and injecting ~500 conidia (in 5 μL of a 10^5^ conidia/mL suspension) into the hemocoel of each larva in each group for cuticle-bypassing infection (CBI). The used conidia were collected directly from the SDAY cultures or impaired at the UVB dose of 0.1 J/cm^2^ after collection. Suspensions of irradiated conidia were prepared by spreading 500 μL aliquots of a 2 × 10^8^ conidia/mL suspension on plates overlaid with cellophane, exposing the plates to the UVB irradiation, resuspending collected conidia in 0.02% Tween 80, and standardizing the suspension to the used concentration. All grouped larvae were held at 25 °C for survival/mortality records every 12 or 24 h. Modeling analysis of the time–mortality trend in each group of larvae infected in either mode was conducted to estimate median lethal time (LT_50_, no. of days) of each strain against the model insect.

### 2.7. Assays for Conidial UVB Resistance and Photoreactivation Rates

Conidial UVB resistances of Δ*rad1*, Δ*rad10*, and control strains were assayed as described previously [6,7]. Briefly, 100 μL aliquots of conidial suspension were evenly smeared onto the plates of germination medium (GM; 2% sucrose, 0.5% peptone, and 1.5% agar). After 10 min drying via sterile ventilation, the plates were irradiated for less than 4 min [6] in the sample tray of a Bio-Sun^++^ UV chamber (Vilber Lourmat, Marne-la-Vallée, France) at the gradient UVB doses of 0.02, 0.03, 0.07, 0.1, 0.15, 0.2, 0.25, 0.3, 0.35, 0.4, and 0.5 J/cm^2^. An error of each dose with weighted UVB wavelength of 312 nm was controlled to ≤1 μJ/cm^2^ (10^–6^) by an inset microprocessor, which automatically adjusted the irradiating wavelength and the intensity four times per second (per the manufacturer’s guide). The irradiated plates were covered immediately with lids and incubated at 25 °C for 24 h in the dark. From 8 h dark incubation onwards, three fields of microscopic view per plate were microscopically examined every 2 h to gain the maximal germination rate of conidia irradiated at each UVB dose. Conidial survival indices (*I*_s_, ratios of maximal germination percentages of irradiated versus non-irradiated conidia) over the UVB doses (*d*) fit the equation *I*_s_ = 1/[1 + exp(*a* + *rd*)] for estimation of parameters *a* and *r*. Lethal doses to inactivate 50% (LD_50_), 75% (LD_75_), and 95% (LD_95_) of tested conidia were computed as indices of conidial UVB resistance by solving the fitted equation at the respective *I*_s_ values of 0.5, 0.25, and 0.05.

Further, the conidia of each strain smeared onto GM plates as above were differentially impaired at the UVB doses of 0.15, 0.3,and 0.4 J/cm^2^ in the Bio-Sun chamber. The irradiated plates were immediately incubated at 25 °C for the optimal 5 h [44] under white light plus 19 h in full darkness (photoreactivation) or directly for 40 h in the dark (NER). Maximal germination rate in each treatment was monitored as mentioned.

### 2.8. Assays for CPD and 6-4PP DNA Lesions in UVB-Impaired Cells

For each of the Δ*wc1* and Δ*wc2* mutants and their control strains, three 50 mL aliquots of a 10^6^ conidia/mL SDBY were shaken (150 rpm) at 25 °C for 3 days. Cells from each culture were rinsed with sterile water and resuspended in 50 mL sterile water. The aliquots of 500 μL suspension were spread on cellophane-overlaid agar plates containing 0.1% yeast extract and exposed to a UVB dose of 0.4 J/cm^2^ for generation of intracellular CPD and 6-4PP DNA lesions. The irradiated plates were covered with lids and incubated at 25 °C for 5 h under white light (photorepair) or in the dark (NER). A Biospin Fungus Genomic DNA Extraction Kit (Bioer Technology, Hangzhou, China) was used to extract genomic DNAs from the cells incubated under light and in the dark or directly from the irradiated cells not incubated (control). Anti-CPD and anti-6-4PP antibodies (Cosmo Bio Co. Ltd., Japan) were used in enzyme-linked immunosorbent assays (ELISAs) to quantify CPD and 6-4PP lesions accumulated in each DNA sample, respectively. Every 20 or 200 ng DNA was added to each well of 50 μL diluent system in a 96-well plate for quantification of CPD or 6-4P PDNA lesions following the manufacturer’s guide. The optical density (OD_492_) of each DNA dilution at 492 nm was used as an index of CPD or 6-4PP DNA lesions accumulated in the irradiated cells before and after photorepair or NER. Three independent DNA samples per strain were included in each treatment.

### 2.9. Transcriptional Profiling

Total RNAs were extracted with a RNAiso Plus Kit (TaKaRa, Dalian, China) from three 3-day-old cultures of each strain grown on cellophane-overlaid SDAY plates at the optimal regime and reversely transcribed into cDNAs with a PrimeScript RT reagent kit (TaKaRa). The cDNA samples were used as templates to assess: (1) transcript levels of *rad1*, *rad10*, *wc1*, or *wc2* in targeted gene mutants and WT for verification of targeted gene recombination events; and (2) transcript levels of 22 DNA damage repair-required genes in Δ*wc1*, Δ*wc2*, and their control strains. The qPCR analysis withpaired primers (Appendix A) was performed under the action of SYBRPremix *ExTaq* (TaKaRa). The fungal β-actin gene was used as a reference. Relative transcript levels of analyzed genes in the mutants were computed with respect to the WT standard using a threshold-cycle (2^−ΔΔCt^) method.

### 2.10. Statistical Analysis

Mean N/C-GFI ratios between Rad1-GFP and Rad10-GFP were compared with Student’s *t* test. All experimental data were subjected to one-factor analysis of variance and Tukey’s honestly significant difference (HSD) tests for differences among tested strains.

## 3. Results

### 3.1. Domain Architecture, Localization, and Interaction of Rad1 and Rad10

The Rad1 and Rad10 orthologues found in selected fungi were clustered to phylogenetic clades or subclades generally associated with their host lineages (Appendix A). The *B. bassiana* Rad1 (EJP63355) and Rad10 (EJP67637) were revealed to share higher sequence identities with the counterparts of entomopathogenic fungi (86–96% and 65–82%) than of non-entomopathogenic fungi (34–84% and 33–59%). Either Rad1 or Rad10 is different in molecular size between *B. bassiana* and *S. cerevisiae* but similar in domain architecture, as shown by the same size of ERCC4 domain (81 aa) and the similar size of Rad10 domain (114 or 118 aa), respectively (Figure 1A). A same size of NLS motif was also predicted from the sequences of their Rad1 (10 aa) or Rad10 (31 aa) orthologues.

The Rad1-GFP and Rad10-GFP fusion proteins expressed in the WT strain accumulated more in the nuclei than in the cytoplasm of the hyphal cells, as shown in LSCM images (Figure 1B). The mean (±SD) N/C-GFI ratios revealed significantly more accumulation of Rad1-GFP (5.3 ± 2.4) than of Rad10-GFP (3.4 ± 1.7) in the nuclei (Figure 1C). Moreover, GFP-tagged Rad1 and mCherry-tagged Rad10 fusion proteins co-expressed in WT merged very well in both the nuclei and cytoplasm out of hyphal vacuoles (Figure 1D), implying a Rad1–Rad10 interaction. This implication was clarified in the Y2H assay. The constructed diploids AD-Rad1-BD-Rad10 and AD-Rad10-BD-Rad1 grew well on the quadruple-dropout plate like the positive control, whereas all negative controls (empty AD or BD) grew only on the double-dropout plate (Figure 1E). This suggests a strong interaction between Rad1 and Rad10 in *B. bassiana* as seen in *S. cerevisiae* [54,55].

### 3.2. Rad1 and Rad10 Are Essential for Preventing DNA Damage but Dispensable for Asexual Cycle In Vitro

Disruption of *rad1* or *rad10* in WT resulted in abolished growth on the SDAY and CDA plates irradiated at the UVB dose of 0.1 J/cm^2^ after inoculation (Figure 2A). Moreover, the mutants became hypersensitive to DNA damage induced by methyl methanesulfonate and mitomycin C (Figure 2B,C). 

In the standardized bioassays, NCI and CBI resulted in similar LT_50_ (±SD) estimates indicative of virulence (5.28 ± 0.30 days via NCI, 3.92 ± 0.18 days via CBI, *n* = 15) for all tested strains (Figure 2D). In contrast, the mutants’ virulence was greatly attenuated by inoculation with conidia impaired at 0.1 J/cm^2^ for NCI or CBI (Figure 2E). The LT_50_ means via NCI and CBI were 5.50 (±0.24) and 4.56 (±0.15) days (*n* = 9) for the three control strains but prolonged to 8.02 (±0.37) and 6.60 (±0.53) days (*n* =3) for Δ*rad1* and 8.98 (±0.30) and 8.20 (±0.18) days (*n* = 3) for Δ*rad10* (Figure 2F), respectively. The mutants’ hypersensitivity to DNA-damaging UVB and chemical agents indicated essential roles for Rad1 and Rad10 in preventing *B. bassiana* from DNA damage.

Compared to the control strains, however, the Δ*rad1* and Δ*rad10* mutants showed insignificant or marginal changes in radial growth on different media (Appendix A) and on CDA or SDAY under osmotic, oxidative, cell perturbing, and thermal stresses (Appendix A). Nor did they display any defect in conidiation at the optimal regime (Appendix A) or in conidial viability assessed as GT_50_ at 25 °C (Appendix A). Apparently, Rad1 and Rad10 were dispensable for the normal asexual cycle in vitro of *B. bassiana*.

### 3.3. Rad1 and Rad10 Have High Photoreactivation, but Insufficient NER, Activities

Modeling analyses of conidial survival trends over gradient UVB doses (Figure 3A) resulted in a mean LD_50_, LD_75_, and LD_95_ of 0.218 (±0.006), 0.275 (±0.008), and 0.370 (±0.012) J/cm^2^ (*n* = 9) for the control strains’ conidia (Figure 3B), respectively. Compared to these means, LD_50_, LD_75_, and LD_95_ were lowered by 79%, 73%, and 67% in Δ*rad1* and 80%, 74%, and 68% in Δ*rad10*, respectively.

Moreover, the control strains’ conidia irradiated at the UVB doses of 0.15, 0.3, and 0.4 J/cm^2^ were photoreactivated by 100%, 95.8% (±3.9), and 87.1% (±2.5), respectively, via 5 h light plus 19 h dark incubation at 25 °C (Figure 3C,D). The photoreactivation rates of conidia irradiated at the gradient doses diminished to 25.3%, 13.3%,and 3.3% for Δ*rad1* and 24%,14%, and 4.3% for Δ*rad10*, respectively. However, the control strains’ conidia exposed to the high dose of 0.4 J/cm^2^ had no germination at all after a 24h dark incubation for NER (Figure 3A,C) but were reactivated by 26.9% (±4.7) (*n* = 9) when the dark incubation was prolonged to 40 h (Figure 3C,E). The 40h dark incubation also resulted in 100% and 51.1% (±4.6) germination of the control strains’ conidia exposed to 0.15 and 0.3 J/cm^2^, respectively, but only ~3% germination of the mutants’ conidia impaired at the low dose (Figure 3E)

The UVB LD_50_ was reduced by 79% or 80% in the Δ*rad1* or Δ*rad10* mutant relative to the control strains. This reduction was much greater than those observed in the previous Δ*phr1* (38%) and Δ*phr2* (19%) mutants [44], highlighting the greater role of either Rad1 or Rad10 than of either photolyase in conidial UVB resistance and also revealing an existance of NER activity in Rad1 or Rad10. The photoreactivation rates of the Δ*rad1* and Δ*rad10* mutants’ conidia inactivated at 0.4 J/cm^2^ were lower or much lower than those seen in the previous Δ*phr1* (7%) and Δ*phr2* (37%) mutants’ conidia inactivated at 0.5 J/cm^2^ [44]. These data demonstrated that Rad1 and Rad10 played essential roles in the photoreactivation of UVB-impaired or UVB-inactivated conidia. Their anti-UVB roles and photoreactivation activities were both greater than those of Phr1 and Phr2 characterized previously in *B. bassiana*. The NER activity of either Rad1 or Rad10 was observed in the control strains’conidia at the end of 24h dark incubation after exposure to ≤0.35 J/cm^2^ but not detectable after exposure to 0.4 J/cm^2^ unless the dark incubation exceeded 24 h, which is far beyond the nighttime of a circadian day. Therefore, Rad1 and Rad10 protect *B. bassiana* from solar UVB damage by photoreactivation. Their NER activities were seemingly extant but hardly sufficient for reactivation of UVB-imapred *B. bassiana* conidia under field conditions due to too short night (dark) time.

### 3.4. Interactions of Rad1 and Rad10 with WC1 and WC2 as Photolyase Regulators

Photoreactivation relies on the photorepair of UVB-induced CPD and 6-4PP DNA lesions by Phr1 and Phr2 in *B. bassiana* [44] or by WC1 and WC2 that interact with both Phr1 and Phr2 in *M. robertsii* [49]. Thus, Y2H assays were carried out to reveal the possible links of Rad1 and Rad10 to WC1/2 or Phr1/2 required for DNA photorepair. In the assays, either Rad1 or Rad10 was proven to interact with both WC1 and WC2 (Figure 4A) but to not interact with Phr1 or Phr2 (Appendix A–D). There was no sign of an interaction between WC1 and Phr1 or Phr2 (Appendix A) or between WC2 and Phr1 or Phr2 (Appendix A). Subsequent Y1H assays revealed the activity of either WC1 or WC2 binding to the promoter regions of both *phr1* and *phr2* (Figure 4B). The detected DNA-binding activities suggest that WC1 and WC2 serve as regulators of two DNA photorepair-dependent photolyases.

Next, ELISAs with anti-CPD and anti-6-4PP antibodies were conducted to assess the accumulated CPD and 6-4PP lesions in the DNA samples isolated from irradiated hyphal cells of Δ*wc1* and Δ*wc2* mutants. The UVB irradiation at 0.4 J/cm^2^ resulted in similar levels of CPD lesions (control in Figure 4C) and differential levels of 6-4PP lesions (control in Figure 4D) in the DNA samples of Δ*wc1*, Δ*wc2*, and their control strains. After 5h light exposure for photorepair, the amount of CPD and 6-4PP lesions were lowered by 34–44% and 30–43% in the control strains’ cells, respectively. However, such photorepair led to insignificant changes in CPD lesions and slight decreases in 6-4PP lesions in the Δ*wc1* and Δ*wc2* mutants’ cells. After 5h dark incubation for NER, CPD lesions decreased in the DNA samples of the control strains slightly more than of Δ*wc1* and Δ*wc2*, while 6-4PP lesions were moderately reduced in all tested strains except Δ*wc2*, whose 6-4PP level was not affected in comparison to the control. These data demonstrate that, in *B. bassiana*, the role of either WC1 or WC2 in photorepairing both CPD and 6-4PP DNA lesions is beyond that of Phr1 specific to CPD or Phr2 specific to 6-4PP in *B. bassiana* [44] and of WC2 specific to CPD or WC1 specific to 6-4PP in *M. robertsii* [49].

The role of WC1 or WC2 in the transcriptional activation of *phr1* and *ph2* was verified by the nearly abolished expression of both *phr1* and *phr2* in either Δ*wc1* or Δ*wc2* (Figure 4E), providing answers to why the two mutants were severely impaired in their capability of photorepairing both CPD and 6-4PP DNA lesions. Moreover, half of the other 20 genes homologous to those required for the yeast NER [27] were markedly repressed by 50–90% (1- to 10-fold) in Δ*wc1* or Δ*wc2*, suggesting active roles of WC1 and WC2 in transcriptional mediation of multiple anti-UV genes in *B. bassiana*.

## 4. Discussion

In *B. bassiana*, Rad1 and Rad10 were proven to co-localize in both the nucleus and the cytoplasm for nucleocytoplasmic shuttling and interact with each other as elucidated in *S. cerevisiae* [53,54]. The extraordinary anti-UV roles of Rad1 and Rad10 rely on their high photoreactivation activities and hence are distinguished from the dependence of the budding yeast orthologues’ anti-UV roles on NER [33,34,35,36]. The photoreactivation activities of Rad1 and Rad10 are speculated to arise from the interactions of each with both WC1 and WC2, which were proven to regulate expression of *phr1* and *phr2* and enable the photorepair of both CPD and 6-4PP DNA lesions, although their interactions with both photolyases were not evidenced in *B. bassiana* as shown previously in *M. robertsii* [49]. At the transcriptional level, importantly, both *phr1* and *phr2* were nearly abolished in the Δ*wc1* or Δ*wc2* mutant. This is different from the abolished expression of *phr2* in the absence of *wc1* and of *phr1* in the absence of *wc2* in *M. robertsii* [49]. Thus, the present Δ*wc1* and Δ*wc2* mutants were severely compromised in photorepair activity for both CDP and 6-4PP lesions, unlike the previous *M. robertstii* Δ*wc1* and Δ*wc2* mutants that were unable to photorepair 6-4PP and CPD DNA lesions, respectively [49]. The present and previous studies suggest that WC1 and WC2 play key roles in the photorepair of UVB- induced DNA lesions no matter whether each acts as a mediator of *phr1* and *phr2* in *B. bassiana* or interacts with both Phr1 and Phr2 in *M. robertsii*. The protein–protein interactions detected in this study suggest direct links of either Rad1or Rad10 to both WC1 and WC2 as photolyase regulators. Therefore, either Rad1 or Rad10 tied to two photolyase regulators displayed much higher UVB resistance and photoreactivation activity than a single photolyase or photolyase mediator in *B. bassiana*, although the involved mechanism remains to be explored.

In the present study, Rad1 and Rad10 did exhibit NER activities for UVB-impaired *B. bassiana* cells like those of their yeast orthologues for UVC-impaired cells [33,34,35,36]. This is well demonstrated by the differential reactivation rates of the control strains’ conidia incubated for 24–40 h in the dark after exposure to the UVB doses of ≤0.35 J/cm^2^ and the abolished germination of the Δ*rad1* and Δ*rad10* mutants’ conidia incubated in the same fashion after exposure to ≤0.15 J/cm^2^. However, the control strains’ conidia impaired at the UVB dose of 0.4 J/cm^2^ were unable to be reactivated within 24 h of dark incubation, in contrast to their high photoreactivation rates of ~87%. This highlights an insufficient NER activity for either Rad1 or Rad10 in the field, where a much shorter nighttime is available for the reactivation of formulated conidia exposed to solar UVB irradiation that accumulates to ~2.5 J/cm^2^ during the daytime [10]. Therefore, either Rad1 or Rad10 tied to two photolyase regulators protects *B. bassiana* from solar UVB damage mainly by photoreactivation.

In conclusion, Rad1 and Rad10 play essential roles in *B. bassiana*’s response and adaptation to solar UV irradiation in sunlight. The essential roles of Rad1 and Rad10 in preventing insecticidal fungal cells from solar UVB damage rely upon their high photoreactivation activities, whichcould have arisen from interactions of either with both WC1 and WC2 to govern photorepair via thetranscriptional mediation of both *phr1* and *phr2*, and hence are distinguished from a dependence of the yeast orthologues’ anti-UV roles on NER. The NER activity of Rad1 or Rad10 is extant in *B. bassiana*, as known in the budding yeast [27], but hardlysufficientin the field where the dark (night) time available for NER is too short. These findings demonstrate a novel scenario for Rad1 and Rad10 to protect filamentous fungal cells from solar UV damage and provide robust evidence for the hypothesis regarding the dependence of filamentous fungal adaptation to solar UV irradiation on the WCC-cored pathway that comprises not only one or two photolyases but also multiple anti-UV RAD proteins [19], which warrants further study. However, Rad1 and Rad10 have no other role in the lifecycle in vitro and in vivo of *B. bassiana*.

## Figures and Tables

**Figure 1 jof-08-01124-f001:**
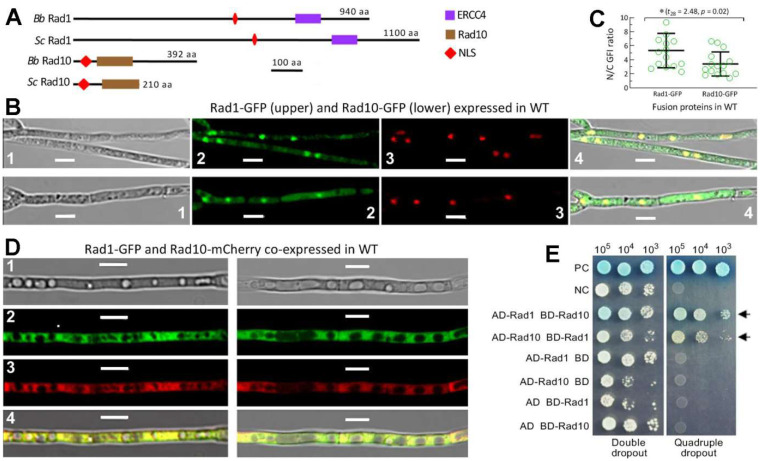
Domain architecture, subcellular localization, and interaction of Rad1 and Rad10 in *B. bassiana*. (**A**) Comparison of conserved domains and NLS motifs predicted from the amino acid sequences of Rad1 and Rad10 orthologues in *B. bassiana* (*Bb*) and *S. cerevisiae* (*Sc*). (**B**) LSCM images (scale bars: 5 μm) for subcellular localization of Rad1-GFP and Rad10-GFP fusion proteins expressed in the 3 d-old SDBY cultures of a *Bb* wild-type (WT) strain at 25 °C and L:D 12:12 and stained with the nuclear dye DAPI (shown in red). Images 1–4 are bright, expressed, stained, and merged views of the same microscopic field, respectively. (**C**) The ratios of nuclear versus cytoplasmic green fluorescence intensities (N/C-GPI) of Rad1-GFP and Rad10-GFP assessed from the 15 hyphal cells of the cultures. * *p* = 0.02 in Student’s *t* test. (**D**) LSCM images (scale bars: 5 μm) for subcellular localization of Rad1-GFP (image 2) and Rad10-mCherry (image 3) fusion proteins co-expressed in the WT strain. (**E**) Y2H assay for an interaction between *Bb* Rad1 and *Bb* Rad10. The diploids AD-Rad1-BD-Rad10 and AD-Rad10-BD-Rad1 (arrowed) grew well like the positive control (AD-LargeT-BD-P53) on the quadruple-dropout medium during a 3-day incubation at 30 °C after initiation of colony growth by 10^3^, 10^4^, and 10^5^ cells.

**Figure 2 jof-08-01124-f002:**
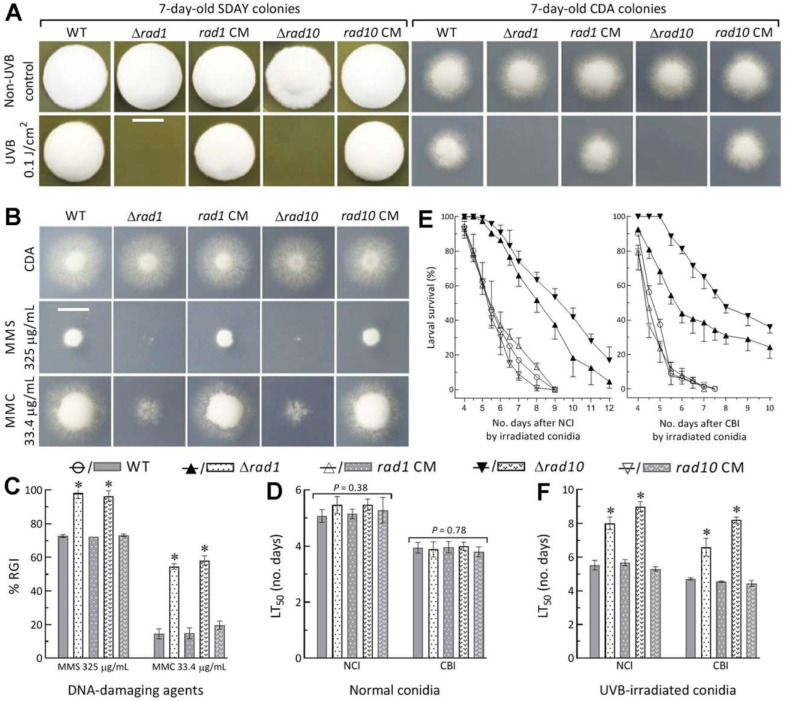
Essential roles of Rad1 and Rad10 in preventing *B. bassiana* from DNA damage. (**A**) Colony images (scale bar: 10 mm) of Δ*rad1*, Δ*rad10*, and their control (WT and CM) strains incubated at 25 °C and L:D 12:12 for 7 days on SDAY and CDA plates irradiated at the UVB dose of 0.1 J/cm^2^ after initiation of colony growth. (**B**,**C**) Colony images (scale bar: 10 mm) and relative growth inhibition (RGI) of the strains incubated at the same regime for 7 days on CDA plates containing the indicated concentrations of methyl methanesulfonate (MMS) and mitomycin C (MMC). All colonies in (**A**–**C**) were initiated with ~10^3^ conidia. (**D**) LT_50_ estimates for tested strains against *G. mellonella* larvae inoculated with normal conidia for normal cuticle infection (NCI) and cuticle-bypassing infection (CBI). (**E**,**F**) Larval survival trends and LT_50_ estimates made from the trends via NCI and CBI initiated with the tested strains’conidia irradiated at the UVB dose of 0.1 J/cm^2^. * *p* < 0.01 (**C**,**F**) in Tukey’s HSD tests. Error bars in (**C**–**F**): standard deviations (SDs) from three independent replicates.

**Figure 3 jof-08-01124-f003:**
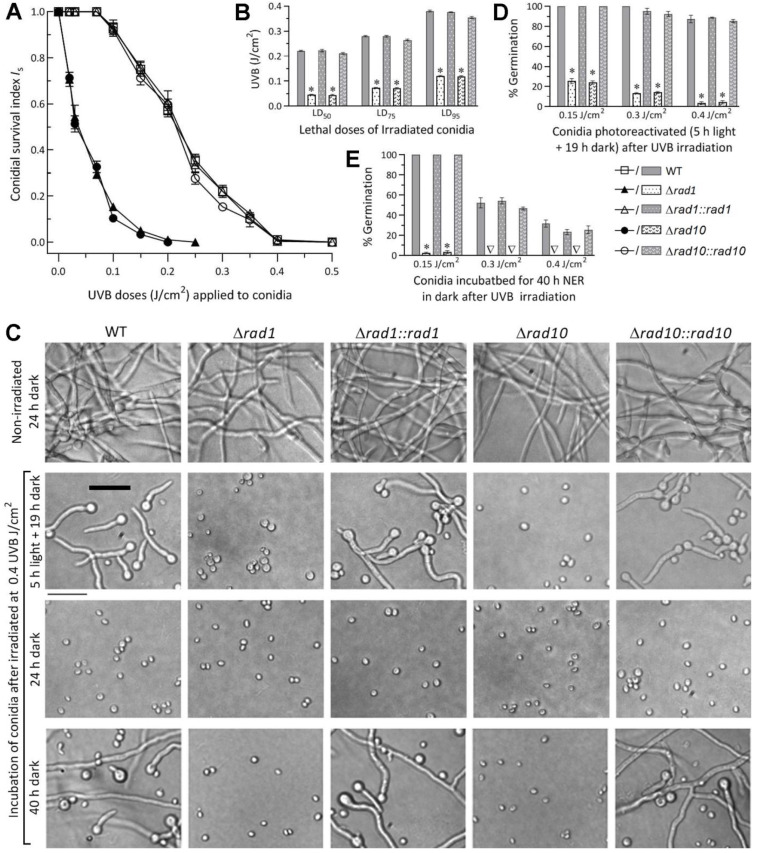
Essential roles of Rad1 and Rad10 in conidial resistance to UVB damage and photoreactivation of UBV-impaired conidia in *B. bassiana*. (**A**) Conidial survival trends of Δ*rad1*, Δ*rad10*, and their control strains over the gradient UVB doses. (**B**) LD_50_, LD_75_, and LD_95_ estimated as indices of conidial UVB resistance by modeling analyses of tested strains’ survival trends. (**C**–**E**) Microscopic images (scale: 20 μm) for germination status and germination percentages of the tested strains’ conidia during an incubation at 25 °C of 5 h under white light plus 19 h in full dark (photoreactivation) or of 40 h in the dark (NER) after they were impaired differentially at the indicated UVB doses. * *p* < 0.0001 in Tukey’s HSD tests. Error bars: SDs from three independent replicates.

**Figure 4 jof-08-01124-f004:**
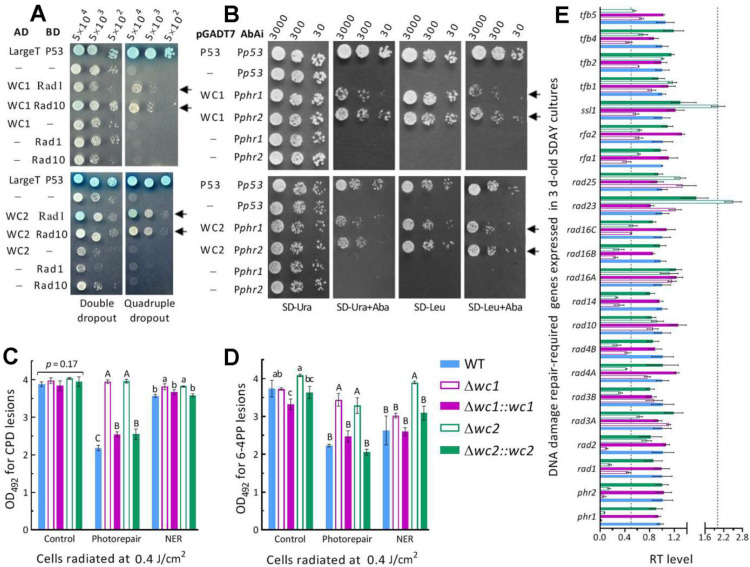
Rad1 and Rad10 interact with WC1 and WC2 serving as regulators of Phr1 and Phr2 in *B. bassiana*. (**A**) Y2H assays for interactions of either Rad1 or Rad2 with both WC1 and WC2. Note that the constructed diploids (arrowed) grew well like the positive control, AD-LargeT-BD-P53, on the quadruple-dropout medium during a 3-day incubation at 30 °C after initiation of colony growth by the indicated numbers of cells. (**B**) Y1H assays for the activities of WC1 and WC2 binding to the DNA fragments of P*phr1* and P*phr2*. Note that the arrowed constructs grown on different media at 30 °C indicate positive binding activities of WC1 or WC2 to both P*phr1* and P*phr2*. (**C**,**D**) The OD_492_ readings for the amounts of CPD and 6-4PP DNA lesions accumulated in the hyphal cells of Δ*wc1*, Δ*wc2*, and their control strains incubated at 25 °C for 5 h under white light (photorepair) and in the dark (NER), respectively, or not incubated (control) after impaired at the UVB dose of 0.4 J/cm^2^. The CPD and 6-4PP OD_492_ values were read from 50 μL reaction systems containing 20 and 200 ng DNA samples probed by anti-CPD and anti-6-4PP antibodies in a 96-well ELISA plate, respectively. Different lowercase or uppercase letters in each bar group denote significant differences of *p* < 0.05 or 0.01 (Tukey’s HSD tests). (**E**) Relative transcript (RT) levels of *phr1*, *phr2*, and 20 putatively NER-required genes in the 3-day-old SDAY cultures of *wc1* and *wc2* mutants with respect to the WT standard. Dashed lines indicate significant levels of one-fold down- and upregulation. Error bars: SDs from three independent replicates.

## Data Availability

All experimental data are included in this paper and Appendix A.

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
