# Peer review of "Rad1 and Rad10 Tied to Photolyase Regulators Protect Insecticidal Fungal Cells from Solar UV Damage by Photoreactivation"

_jof, 2022, doi:10.3390/jof8111124_

Round 1

Reviewer 1 Report

The study shows an important anti-UV radiation role of Rad1 and Rad10 of B. bassiana by photoreactivation instead of by the nucleotide excision repair mechanism exhibited by their orthologue proteins in yeasts. The authors suggest that photoreactivation activities of both Rad1 and Rad10 are due to interactions of each with both WC1 and WC2, members of the white-collar complex, as conclusion of a yeast hybrid assay. However, it is not clear why the transcriptomic profiling of the potential damage repair-required genes was done in both Wc mutants but not in both Rad mutants. This seems to be related with previous studies of the authors, but it is difficult to understand in this piece of research. Please explain. How were these potential damage repair-required genes chosen? Also, why was the transcriptomic analysis not done by RNAseq? This would have expanded the transcriptomic profile to have stronger conclusions.

Section 2.7: Please detail what is a gradient UV-B doses. How many doses were assayed? What was the irradiance received by each sample and for how long to obtain the indicated UV-B doses?

Author Response

Please see attached a file for author response to review.

Reviewer 2 Report

In this work, Yu and colleagues describe the involvement of endonuclease complex Rad1-Rad10 from the filamentous fungus Beauveria bassiana along with the photolyase regulator in UV protection. The authors generated the deletion mutant for both Rad1-Rad10, and tested their roles in anti-UV and asexual growths in detail. Furthermore, the authors detected the direct interaction between these two Rad proteins and two other white-collar proteins (WC1 and WC2). Additionally, the transcriptional regulation between WC1-WC2 and photolyases PHR1-PHR2 was also determined. Overall, this is an interesting paper for the scientists working on DNA repair and UV protection in the filamentous fungus, but I feel the logical connection and throughout thinking are missing in the current manuscript. It is hard for the readers to realize why and how some experiments were performed. Therefore, there are some issues with the work presented that need to be addressed before the manuscript might be considered acceptable. 

Major concerns

1. The result is interesting, but the logical connection is weak. For example, I am not sure why the author suddenly studied the subcellular and co-localization of Rad1 and Rad10? Please tell the readers the reason and logic for the upcoming assay before describing the outcome of each experiment.  

2. Line 392-395, The authors concluded that these two proteins are “Infeasible” in NER, which is in doubt for me. If the germination rate at both 24-dark incubation and 40-dark incubation were considered as the experimental evidence for the gene function in NER, then the results obviously suggest that Rad1 and Rad10 are important for NER since there is only ~3 % of germination for these two mutants at 40h. In terms of the data from 24h, 0 germination rate for both wildtype (baseline in this assay) and mutants only suggests that the overall NER is absent in this scenario, it is hard to interpret the function of Rad1-Rad10 in this setting. Therefore, I would recommend the authors perform a new experiment that treats the fungi with a weaker strength of UV (e.g., 0.15 J/cm2 or 0.3 J/cm2) prior to counting the germination rate at 24 h dark. I assume that in this case, there will be at least some wildtype surviving, which would be considered as baseline and standard to determine the impact of Rad1-10 knockout.

3. Line 429-430, The authors claimed the importance of WC1 and WC2 in the transcriptional regulation of multiple anti-UV genes. I am cautious that since the sample used in the RNA-seq condition is from a normal growth condition (SDAY plates) rather than a UV-stressed condition, how interpretable the data is for UV protection?  Is the UV protection network keeping working even without UV stress? qPCR for some rad proteins and phr1/2 from UV irradiated sample would be helpful to confirm the role of WC1/2 in anti-UV regulation if a new RNA-seq assay is not feasible.

4. Line454-456, I don’t think the author provided strong evidence to support their conclusion about “The photoreactivation activities of Rad1 and 454 Rad10 arise from interactions of each with both WC1 and WC2”. Their interaction is supported by Y2H assay for sure, but how do you know this interaction is crucial for their photoactivation activities? Therefore, this reviewer feels that the connection between the first (Rad1-10 in photorepair) and the second part of the result (WC1-2 and PHR1-2 in photorepair) is weak.

5. The abstract does not follow the order of presentation in the result section. Please re-organize it.

Minor concerns

Line 20, rad1 and rad10 should be replaced with RAD1 and RAD10.

Line 33-34, “understanding the roles and mechanisms behind fungal anti-UV”.

Line 60, the inserted place for citations 15 and 26 is weird.

Line 62, please be more specific, budding yeast. Same for the rest.

Line 59, a more detailed introduction for NER is needed.

Line 73, Rad1-Rad10 is also important for microhomology mediated end-joining (MMEJ) single strand annealing (SSA), please include that and see details in Huang and Cook., 2022. DOI: 10.1093/femsre/fuac035.

Line 74, I would like to suggest the authors tune down their statement about “has not been studied yet in filamentous fungi”, please see the study from Neurospora crassa, Hatakeyama et al., 1998. DOI: 10.1007/s002940050337.

Line 76-80, should be moved to be the discussion

Line 84, to not or does not?

Line 86, and required.

Line 85-90, this sentence is too long.

Line90, it is supposed to be “Neurospora crassa”, please fix that typo.

Line 94-95, “A circadian day of summer features long day- 94 time and short nighttime insufficient for NER in the field” is confusing, please re-organize it.

Line291, Filamentous fungal is a weird phrase, I would like to suggest the authors use “xxx from filamentous fungi”

Line293, Figure-S2, many proteins are not from filamentous fungi (e.g., Candida and Ustilago), please re-phrase.

Line 315, It’s not clear to me why the authors want to check subcellular localization? Additionally, I found the localization pattern for Rad1-GFP is different in Fig1B and Fig1D, the nuclear localization seems much less in FIG1D, any explanation?

Line 318, Rad10-GFP

Line313, please also clarify negative control, as you did for positive control

Line328-339, I did not find any descriptions for the CM strains in the result. Same issue for all the complemented strains in this manuscript.

Line358, the normal fungal asexual cycle

Line 362, “control strains' conidia (Figure 3B)” means wildtype? or complemented strain?

Line 377, 25.3% 24%, and 13.3%; Line 378, 14%, 3.3%, and 4.3%, these numbers do not make sense to me, and do not match the data presented in Fig3D.

Line 378-382, the sentence is confusing, please re-phrase it.

Line 369, what do the transparent triangles mean in Fig3E (middle and right panel)

Line 384, LD50s? what does “s” mean here? Also, delete “the” before 79%.

Line 411, please add a delta sign “Δ “  before wc1 and wc2

Line 444, it is confusing to me why the authors use both small and capital letters.

Line 420-423, I am confused about this conclusion, please re-phrase it. Instead of saying your result is different than others, why not highlight what you found?

Line 428-429, “other 20 genes homologous to those required for the yeast NER [27] were significantly repressed by 50–90%”. Please provide the statistical analysis to support the statement about the significance.

Author Response

(The authors gave the same response as above.)

Round 2

Reviewer 1 Report

All comments were answered

Author Response

I am pleased to see that you are satisfied with our revision and responses to your comments and suggestions. Thank you very much for reviewing our manuscript.

Reviewer 2 Report

In this revision, the authors addressed some of my comments. However, I am disappointed with the rest of their responses, and the unwillingness of the authors to take my comments seriously.

Major comments

Major comments#1: Please briefly summarize and add your explanation to my major concern#1 to the manuscript.

Major comments#2: Okay, the response from the authors “The suggested experiments were exactly the same as shown in Figure 2C-E.”

Let’s see what I suggested “Therefore, I would recommend the authors perform a new experiment that treats the fungi with a weaker strength of UV (e.g., 0.15 J/cm2 or 0.3 J/cm2) prior to counting the germination rate at 24 h dark. I assume that in this case, there will be at least some wildtype surviving, which would be considered as baseline and standard to determine the impact of Rad1-10 knockout.”

Then what the authors did in 2C-E

2C: “Also, the mutants became hypersensitive to DNA damage induced by methyl methanesulfonate and mitomycin C (Figure 2B and C)”  

-------Is this assay related to counting the germination rate of UV-treated conidia after 24 h dark (NER)? For me, the answer is NO.

2D: “In the standardized bioassays, NCI and CBI resulted in similar LT50 (±SD) estimates indicative of virulence (5.28±0.30 days via NCI, 3.92±0.18 330 days via CBI, n=15) for all tested strains (Figure 2D).”

-------Is this assay related to counting the germination rate of UV-treated conidia after 24 h dark (NER)? For me, the answer is NO.

2E:  In contrast, the mutants' virulence was greatly attenuated by inoculation with conidia impaired at 0.1 J cm–2 for NCI or CBI (Figure 2E).”

-------Is this assay related to counting the germination rate of UV-treated conidia after 24 h dark (NER)? For me, the answer is NO.

In my opinion, the authors did not provide convincing data to support “Rad1-10 have infeasible NER activities”. If you say NER activities are not feasible, I could somehow agree, but If you say NER activities of Rad1-10 are not feasible, please provide the data.

Major comments#5:  YES, IT IS NECESSARY. Disordered abstract definitely impairs readability.

Minor comments

1. “Line 76-80, should be moved to be the discussion
Author response:”

--I do not see your response

2. Line 444, it is confusing to me why the authors use both small and capital letters.
--Author response: Small and capital letters in line 444? I am confused too.

My question is why you chose distinct signs (lowercase or uppercase letters) to indicate the significance. Is that common to use different p-values in one single graph? Why not just use p<0.05?  Hope it is clear to the authors now.

3. Line 428-429, “other 20 genes homologous to those required for the yeast NER [27]
were significantly repressed by 50–90%”. Please provide the statistical analysis to
support the statement about the significance.
Author response: Statistical significance is nothing meaningful for transcriptional
changes. We used more strict standard of one-fold change for recognition of up- or
downregulated genes as usually applied in transcriptomic analysis. For example, 5%
or 10% transcript change could be statistically significant but mean nothing for gene
expression.

-- Statistical analysis is REQUIRED for any meaningful transcriptional analysis.  I can understand the author feels one-fold change is stricter than 5% to 10% change. BUT if you want to use the word “significant”, please show the p-value (or indicated by an asterisk)

For your reference

https://journals.plos.org/plospathogens/article?id=10.1371/journal.ppat.1006604

This is an example of how a decent qPCR was analyzed. (From Fig6 Cen et al., 2017, DOI: 10.1371/journal.ppat.1006604, the assay was also done in Beauveria bassiana)

Author Response

Please see attached a file for our point-by-point responses (marked in red) to your second-run comments on our manuscript. We have our revision and responses to have addressed your concerns. Thank you very much for reviewing our manuscript.
